# Efficient Approximate Search for Multi-Objective Multi-Agent Path Finding

## Abstract

The Multi-Objective Multi-Agent Path Finding (MO-MAPF) problem is the problem of computing collision-free paths for a team of agents while considering multiple cost metrics. Most existing MO-MAPF algorithms aim to compute the Pareto frontier of the solutions. However, a Pareto frontier can be time-consuming to compute and contain solutions with similar costs. Our first main contribution is BB-MO-CBS-pex, an approximate MO-MAPF algorithm that computes an approximate frontier for the user-specific approximation factor. BB-MO-CBS-pex builds upon BB-MO-CBS, a state-of-the-art MO-MAPF algorithm, and leverages A*pex, a state-of-the-art single-agent multi-objective search algorithm, to speed up different parts of BB-MO-CBS. We also provide two speed-up techniques for BB-MO-CBS-pex. Our second main contribution is BB-MO-CBS-k, which builds upon BB-MO-CBS-pex and computes up to $k$ solutions for a user-provided $k$-value. BB-MO-CBS-k is useful when it is unclear how to determine an appropriate approximation factor. Our experimental results show that both BB-MO-CBS-pex and BB-MO-CBS-k solved significantly more instances than BB-MO-CBS for different approximation factors and $k$-values, respectively. Additionally, we compare BB-MO-CBS-pex with an approximate baseline algorithm derived from BB-MO-CBS and show that BB-MO-CBS-pex achieved speed-ups up to two orders of magnitude.

## Introduction

The Multi-Agent Path Finding (MAPF) problem is the problem of finding a set of collision-free paths for a team of agents. It is related to many real-world applications (Wurman, D'Andrea, and Mountz 2008; Morris et al. 2016). A *solution* is a set of collision-free paths for all agents. Computing a minimum-cost solution for the MAPF problem is known to be NP-hard (Yu and LaValle 2013; Ma et al. 2016). In this paper, we study a variant of the MAPF problem called the Multi-Objective MAPF (MO-MAPF) problem (Ren, Rathinam, and Choset 2022), which considers multiple cost metrics. Many real-world applications of MAPF can be viewed as multi-objective problems. For example, in multi-robot systems, some interesting cost metrics are travel distance, energy consumption, and risk.

Most existing MO-MAPF algorithms, such as MO-M* (Ren, Rathinam, and Choset 2021), MO-CBS (Ren, Rathinam, and Choset 2022), and BB-MO-CBS (Ren et al.

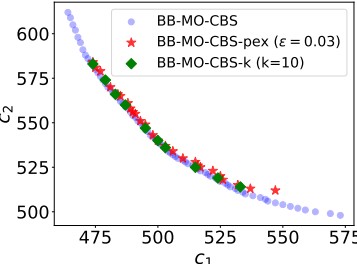

Figure 1: Costs of the solutions computed by different algorithms for an MO-MAPF instance with two objectives and 12 agents, where BB-MO-CBS-pex and BB-MO-CBS-k, our proposed algorithms, achieved speed-ups of $25\times$ and $44\times$ over BB-MO-CBS, respectively.

2023), aim to compute the Pareto frontier of the solutions, that is, a set of all solutions that are not dominated by any other solutions, where a solution $P$ *dominates* another solution $P'$ if the cost of $P$ is no larger than the cost of $P'$ for every cost metric and the cost for at least one cost metric is smaller. Unfortunately, even in the single-agent case, the size of the Pareto frontier can be exponential in the size of the graph being searched (Ehrgott 2005; Breugem, Dollevoet, and van den Heuvel 2017). Therefore, computing Pareto frontiers for MO-MAPF can be time-consuming. Existing works on multi-objective single-agent search have been proposed to compute an approximate frontier (Perny and Spanjaard 2008; Goldin and Salzman 2021; Zhang et al. 2022) instead, which significantly speeds up the search. However, this has yet to be investigated for MO-MAPF.

Our first main contribution is *BB-MO-CBS-pex*, an approximate MO-MAPF algorithm that computes an approximate frontier for the user-specific approximation factor. BB-MO-CBS-pex builds upon BB-MO-CBS, a state-of-the-art MO-MAPF algorithm that consists of a low-level search to plan paths for each agent and a high-level search to resolve collisions. BB-MO-CBS-pex leverages A*pex (Zhang et al. 2022), a state-of-the-art single-agent multi-objective approximate search algorithm, as the low-level search algorithm and also applies the algorithmic idea behind A*pex to to speed up the high-level search. In addition, we provide two techniques to further speed up BB-MO-CBS-pex.

In practice, a too large approximation factor can cause

BB-MO-CBS-pex to return only one solution, which offers no trade-off to users, while a too small one provides no chance for BB-MO-CBS-pex to speed up. Therefore, one might prefer to specify the desired number of solutions instead. Our second main contribution is *BB-MO-CBS-k*, which builds upon BB-MO-CBS-pex and computes a set of up to $k$ solutions for a user-provided $k$-value.

In our experimental study, we compare BB-MO-CBS-pex and BB-MO-CBS-k with BB-MO-CBS. Our results show that BB-MO-CBS-pex and BB-MO-CBS-k solved significantly more problem instances than BB-MO-CBS within the given runtime limit of 120 seconds for different approximation factors and $k$-values, respectively. Additionally, we compare BB-MO-CBS-pex with BB-MO-CBS-$\varepsilon$, an approximate baseline algorithm derived from BB-MO-CBS. Our results show that BB-MO-CBS-pex solved significantly more instances and achieved up to two orders of magnitude speed-up compared to BB-MO-CBS-$\varepsilon$.

## Terminology and Problem Definition

We use **boldface** font to denote vectors or vector functions and $v_i$ to denote the $i$-th component of vector or vector function $\mathbf{v}$. We define the addition of two $M$-dimensional vectors $\mathbf{u}$ and $\mathbf{v}$ as $\mathbf{u}+\mathbf{v} = [u_1+v_1, u_2+v_2, \ldots, u_M+v_M]$. We define the *vector minimum* of $\mathbf{u}$ and $\mathbf{v}$ as $vector\_min(u, v) = [\min(u_1, v_1), \min(u_2, v_2), \ldots, \min(u_M, v_M)]$. $\mathbf{u} \preceq \mathbf{v}$ denotes that $u_i \leq v_i$, $i = 1, 2, \ldots, M$. In this case, we say that $\mathbf{u}$ *weakly dominates* $\mathbf{v}$. $\mathbf{u} \prec \mathbf{v}$ denotes that $\mathbf{u} \preceq \mathbf{v}$ and $\exists i \in \{1, 2, \ldots, M\}$, $u_i < v_i$. In this case, we say that $\mathbf{u}$ *dominates* $\mathbf{v}$. $\mathbf{u} \preceq_{\boldsymbol{\varepsilon}} \mathbf{v}$ for an *approximation factor* (or, more precisely, vector of approximation factors) $\boldsymbol{\varepsilon} = [\varepsilon_1, \varepsilon_2, \ldots, \varepsilon_M]$ denotes that $u_i \leq (1 + \varepsilon_i)v_i$, $i = 1, 2, \ldots, M$. In this case, we say that $\mathbf{u}$ $\boldsymbol{\varepsilon}$-*dominates* $\mathbf{v}$.

In the MO-MAPF problem, we are given a shared workspace, represented by a finite directed graph $G = \langle V, E \rangle$ and, a set of $N$ agents $\{a^1, a^2, \ldots, a^N\}$. $V$ denotes the set of vertices, and each vertex $v \in V$ corresponds to a possible location for agents. $E \subseteq V \times V$ denotes the set of edges, and each edge $e = \langle u, v \rangle \in E$ corresponds to a move action from $u$ to $v$. Note that an edge from a vertex to itself can also be included in $E$, which means that agents can wait at the vertex. The *cost* of an edge $e$ is a positive $M$-dimensional vector denoted as $\mathbf{c}(e) \in \mathbb{R}_{>0}^M$, where $M$ is the *number of objectives*. The agents are indexed by $I = \{1, 2 \ldots N\}$. In the rest of the paper, we use $|I|$ instead of $N$ to denote the number of agents. We use superscript $^{i \in I}$ to indicate that a variable is related to agent $a^i$. Each agent $a^i$ has a *start vertex* $v_{\text{start}}^i \in V$ and a *goal vertex* $v_{\text{goal}}^i \in V$.

A *path* $\pi^i = (v_1^i, v_2^i, \ldots, v_\ell^i)$ for agent $a^i$ is a sequence of vertices with $v_1^i = v_{\text{start}}^i$, $v_\ell^i = v_{\text{goal}}^i$, and $\langle v_j^i, v_{j+1}^i \rangle \in E$, $j = 1, 2 \ldots \ell - 1$. The *cost* of path $\pi^i$ is defined as $\mathbf{c}(\pi^i) = \sum_{j=1}^{\ell-1} \mathbf{c}(\langle v_j^i, v_{j+1}^i \rangle)$. A path also corresponds to a sequence of move and wait actions. We assume that agents stay at their goal vertices forever after they execute their last actions.

For a set of agent indices $I'$, a *joint path* $P = \{\pi^i : i \in I'\}$ is a set of paths, one for each agent whose index is in $I'$. Throughout this paper, we assume that $I = I'$, unless

mentioned otherwise. The *cost* of joint path $P$ is defined as $\mathbf{c}(P) = \sum_{i \in I'} \mathbf{c}(\pi^i)$. We consider two types of conflicts: A *vertex conflict* happens when two agents stay at the same vertex simultaneously, and an *edge conflict* happens when two agents switch their vertices simultaneously. A *solution* is a conflict-free joint path.

In this paper, we use symbol $P$ to denote a joint path, which is a set of paths for different agents, and symbol $\Pi$ to denote a set of paths for the same agent. Additionally, we use symbol $\mathbb{P}$ to denote a set of joint paths.

We say that a path $\pi$ weakly dominates another path $\pi'$ (resp. $\pi$ $\varepsilon$-dominates $\pi'$) if $\mathbf{c}(\pi) \preceq \mathbf{c}(\pi')$ (resp. $\mathbf{c}(\pi) \preceq_{\boldsymbol{\varepsilon}} \mathbf{c}(\pi')$). A set of paths $\Pi$ is *undominated* if its paths do not weakly dominate each other. A *Pareto frontier* of $\Pi$ is defined as an undominated subset of $\Pi$ such that each path in $\Pi$ is weakly dominated by at least one path in the Pareto frontier. An $\varepsilon$-*approximate frontier* for $\Pi$ is defined as an undominated subset of $\Pi$ such that each path in $\Pi$ is $\varepsilon$-dominated by at least one path in the $\varepsilon$-approximate frontier.

For joint paths, we define weakly dominance, $\varepsilon$-dominance, undominated sets, Pareto frontiers, and $\varepsilon$-approximate frontiers in the same way that we do for paths. Unless mentioned otherwise, we use a Pareto frontier (resp. an $\varepsilon$-approximate frontier) to refer to a Pareto frontier (resp. an $\varepsilon$-approximate frontier) of all solutions for the MO-MAPF problem instance we consider.

## Algorithm Background

This section reviews CBS (Sharon et al. 2015), BB-MO-CBS (Ren et al. 2023), and A*pex (Zhang et al. 2022).

### CBS

CBS (Sharon et al. 2015) is a complete and optimal single-objective MAPF algorithm. We omit the pseudocode for CBS due to the space limit. CBS consists of two levels. On the high level, CBS performs a best-first search on a *Constraint Tree* (CT). Each CT node contains (1) a set of constraints and (2) a minimum-cost joint path that satisfies all these constraints. A *constraint* has the form $\langle i, v, t \rangle$ or $\langle i, e, t \rangle$, where $i \in I, v \in V, e \in E, t \in \mathbb{N}_{>0}$. For the first case, any path $\pi^i = (v_1^i, v_2^i, \ldots, v_l^i)$ for $a^i$ is prohibited from $v_t^i = v$; for the second case, any path $\pi^i = (v_1^i, v_2^i, \ldots, v_l^i)$ for $a^i$ is prohibited from $\langle v_t^i, v_{t+1}^i \rangle = e$. The $g$-value of a CT node is defined as the cost of its joint path. CBS maintains an *Open* list for all generated but not yet expanded nodes and initializes *Open* with the root CT node, which has an empty set of constraints and a path for each agent that has the minimum path cost when ignoring conflicts. In each iteration, CBS extracts a CT node with the minimum $g$-value from *Open* and returns its joint path as the solution if the joint path is conflict-free. Otherwise, CBS picks a conflict of the joint path to resolve, splits the CT node into two child CT nodes, and adds a constraint to each child CT node to prohibit either one or the other of the two conflicting agents from using the conflicting vertex or edge at the conflicting timestep. CBS then calls its low level to replan the path of the newly constrained agent in each child CT node. The low level planner finds a path with the minimum path cost while

satisfying all constraints of the child CT node but ignoring conflicts.

## BB-MO-CBS

BB-MO-CBS (Ren et al. 2023) generalizes CBS from single-objective MAPF to MO-MAPF. Given an MO-MAPF problem instance, BB-MO-CBS computes a Pareto frontier of its solutions.

Algorithm 1 shows the pseudocode for BB-MO-CBS. BB-MO-CBS maintains an $Open$ list for all generated but not yet expanded nodes and a solution set $\mathcal{S}$ for the solutions it has found. Similar to CBS, BB-MO-CBS also consists of two levels. On the high level, BB-MO-CBS maintains a CT. A major difference between CBS and BB-MO-CBS is that, while a CT node of CBS corresponds to one joint path, a CT node of BB-MO-CBS corresponds to a set of joint-paths that are different combinations of Pareto-optimal path for each agent. This design allows BB-MO-CBS to resolve the same conflict in different joint paths simultaneously. More specifically, in BB-MO-CBS, we redefine a CT node as a tuple $n = \langle \Omega, \{\Pi^i \mid i \in I\}, \mathbb{P} \rangle$, which contains (1) a set of constraints $\Omega$, where a constraint has the same form with the constraints in CBS, (2) a Pareto frontier of paths $\Pi^i$ for each agent $a^i$ that satisfy constraints in $\Omega$, and (3) a set of joint paths $\mathbb{P} \subseteq PF(\Pi^1 \times \Pi^2 \times \cdots \times \Pi^{|I|})$, where $PF(\Pi^1 \times \Pi^2 \times \ldots \times \Pi^{|I|})$ denotes a Pareto frontier of all joint paths that consist of a path from $\Pi^i$ for each agent $a^i$. As we will show later, BB-MO-CBS repeatedly updates $\mathbb{P}$ to the subset of $PF(\Pi^1 \times \Pi^2 \times \cdots \times \Pi^{|I|})$ that are not weakly dominated by any solution in $\mathcal{S}$. We use $\mathbb{P}.lexFirst$ to denote the joint path with the lexicographically smallest cost in $\mathbb{P}$ and call it the *current joint path* of node $n$. The **g**-value of node $n$ is defined as $\mathbf{c}(\mathbb{P}.lexFirst)$.

During the initialization, BB-MO-CBS first computes a Pareto frontier of paths $\Pi_o^i$, ignoring other agents, for each agent $a^i$, and a Pareto frontier of joint paths $\mathbb{P}_o = PF(\Pi_o^1 \times \Pi_o^2 \times \cdots \times \Pi_o^{|I|})$ (Lines 1-4). It then initializes $Open$ with the root CT node $n_o = \langle \emptyset, \{\Pi_o^i \mid i \in I\}, \mathbb{P}_o \rangle$ (Line 5).

In each iteration, BB-MO-CBS extracts a CT node $n = \langle \Omega, \{\Pi^i \mid i \in I\}, \mathbb{P} \rangle$ with the lexicographically smallest **g**-value (Line 7). The current joint path of $n$, that is, $\mathbb{P}.lexFirst$, must have the lexicographically smallest cost among (and hence is not dominated by) the joint paths of all CT nodes in $Open$. BB-MO-CBS first computes $\mathbb{P}'$ by removing the joint paths weakly dominated by any solution in $\mathcal{S}$ from $\mathbb{P}$. If $\mathbb{P}'$ is empty, BB-MO-CBS discards node $n$ and ends the iteration (Line 9). If the current joint path changes, (that is, $\mathbb{P}'.lexFirst \neq \mathbb{P}.lexFirst$), BB-MO-CBS reinserts a CT node with the updated joint path set $\mathbb{P}'$ to $Open$ and ends this iteration (Lines 10-12). If the current joint path does not change and is conflict-free, BB-MO-CBS adds it to $\mathcal{S}$. Different from CBS, BB-MO-CBS does not terminate in this case. It removes the new solution $\mathbb{P}'.lexFirst$ from $\mathbb{P}'$ and reinserts a CT node with the updated joint path set $\mathbb{P}'$ to $Open$ if $\mathbb{P}'$ is still not empty (Lines 14-19). BB-MO-CBS does this because the remaining joint paths in $\mathbb{P}'$ still have the potential to lead to new solutions. If the current joint path is not conflict-free, similar to CBS, BB-MO-CBS

---

**Algorithm 1** BB-MO-CBS

1: $\mathcal{S} \leftarrow \emptyset; Open \leftarrow \emptyset$
2: **for all** $i \in I$ **do**
3:     $\Pi_o^i \leftarrow LowLevelSearch(i, \emptyset)$
4: $\mathbb{P}_o \leftarrow PF(\Pi_o^1 \times \Pi_o^2 \times \cdots \times \Pi_o^N)$
5: add $n_o = \langle \emptyset, \{\Pi_o^i | i \in I\}, \mathbb{P}_o \rangle$ to $Open$
6: **while** $Open \neq \emptyset$ **do**
7:     $n = (\Omega, \{\Pi^i | i \in I\}, \mathbb{P}) \leftarrow Open.extract()$
8:     $\mathbb{P}' \leftarrow \{P \mid P \in \mathbb{P} \land \nexists P' \in \mathcal{S}\ \mathbf{c}(P') \preceq \mathbf{c}(P)\}$
9:     **if** $\mathbb{P}' = \emptyset$ **then continue**
10:     **if** $\mathbb{P}'.lexFirst \neq \mathbb{P}.lexFirst$ **then**
11:         add $n = \langle \Omega, \{\Pi^i | i \in I\}, \mathbb{P}' \rangle$ to $Open$
12:         **continue**
13:     $cft \leftarrow DetectConflict(\mathbb{P}'.lexFirst)$
14:     **if** $cft = \emptyset$ **then**
15:         add $\mathbb{P}'.lexFirst$ to $\mathcal{S}$
16:         remove $\mathbb{P}'.lexFirst$ from $\mathbb{P}'$
17:         **if** $\mathbb{P}' \neq \emptyset$ **then**
18:             add $\langle \Omega, \{\Pi^i | i \in I\}, \mathbb{P}' \rangle$ to $Open$
19:         **continue**
20:     $\{\omega^i, \omega^j\} \leftarrow GenerateConstraints(cft)$
21:     **for all** $i' \in \{i, j\}$ **do**
22:         $\{\Pi_{new}^i | i \in I\} \leftarrow \{\Pi^i | i \in I\}$
23:         $\Omega_{new} \leftarrow \Omega \cup \{\omega^{i'}\}$
24:         $\Pi_{new}^{i'} \leftarrow LowLevelSearch(i', \Omega_{new})$
25:         $\mathbb{P}_{new} \leftarrow PF(\Pi_{new}^1 \times \Pi_{new}^2 \times \cdots \times \Pi_{new}^N)$
26:         add $\langle \Omega_{new}, \{\Pi_{new}^i | i \in I\}, \mathbb{P}_{new} \rangle$ to $Open$
27: **return** $\mathcal{S}$

---

picks a conflict of the joint path to resolve, splits the CT node into two child CT nodes, and adds a constraint to each child CT node. BB-MO-CBS then calls its low level to replan a Pareto frontier of paths for the newly constrained agent in each child CT node that satisfy all constraints of the child CT node. The low level planner of BB-MO-CBS can be implemented with any single-agent multi-objective search algorithm that computes a Pareto frontier, such as BOA* (Ulloa et al. 2020) and EMOA* (Ren et al. 2022).

BB-MO-CBS terminates and returns $\mathcal{S}$ when $Open$ is empty. Ren et al. (2023) showed that $\mathcal{S}$ is a Pareto frontier for the given MO-MAPF problem instance.

A straightforward approach to introduce approximation to BB-MO-CBS is to prune joint paths that are $\varepsilon$-dominated by any found solution. We propose BB-MO-CBS-$\varepsilon$, an approximate variant of BB-MO-CBS that we will use as a baseline. Given an MO-MAPF problem instance and an $\varepsilon$-value, BB-MO-CBS-$\varepsilon$ computes an $\varepsilon$-approximate frontier of the solutions. BB-MO-CBS-$\varepsilon$ only changes one line of BB-MO-CBS, that is, when computing the updated joint path set $\mathbb{P}'$ (Line 8), it removes all the joint paths in $\mathbb{P}$ that are $\varepsilon$-dominated by any solution in $\mathcal{S}$.

## A*pex

A*pex (Zhang et al. 2022) is a multi-objective (single-agent) search algorithm that computes an $\varepsilon$-approximate frontier of paths from a given start state $v_{\text{start}}$ to a given goal state $v_{\text{goal}}$ for a user-provided $\varepsilon$-value. In A*pex, a node $n$ corresponds to a set of paths $\Pi$ from $v_{start}$ to some vertex $v$. Instead

of explicitly storing $\Pi$, A*pex stores only one path $\pi \in \Pi$, called the *representative path* of $n$, and a cost vector $\mathbf{A}(n)$, called the *apex* of $n$. Cost vector $\mathbf{A}(n)$ is the vector minimum value of the costs of all paths in $\Pi$. We say that node $n$ is $\varepsilon$-*bounded* if $\mathbf{c}(\pi) + \mathbf{h}(v) \preceq_{\varepsilon} \mathbf{A}(n) + \mathbf{h}(v)$, where $\mathbf{h}$ is a consistent heuristic function where each component of $\mathbf{h}(v)$ provides a lower bound on the cost of any path from $v$ to the goal vertex $v_{goal}$ for each objective.

By *merging* nodes whose representative paths end at the same vertex on condition that the resulting node is $\varepsilon$-bounded, A*pex reduces the search effort and can quickly compute an $\varepsilon$-approximate frontier. When merging two nodes $n$ and $n'$, the new apex is the vector minimum of the $\mathbf{A}(n)$ and $\mathbf{A}(n')$, and the new representative path is either one of the two representative paths of $n$ and $n'$. Zhang et al. (2022) proposed several approaches for choosing the new representative path. When expanding a node $n$ that reaches $v_{goal}$, A*pex adds the representative path of $n$, denoted as $\pi$, to the solution set it maintains. Slightly abusing the notation, we use $\mathbf{A}(\pi)$ to denote $\mathbf{A}(n)$ and call it the *apex* of $\pi$. A*pex terminates and returns a set of solutions, denoted as $\Pi_{\varepsilon}$, when its open list becomes empty. In the rest of this paper, we assume that A*pex also outputs the apexes of solutions in $\Pi_{\varepsilon}$. Let $\Pi_{*}$ denote a Pareto frontier from $v_{start}$ to $v_{goal}$. The apexes of paths in $\Pi_{\varepsilon}$ collectively "lower-bound" $\Pi_{*}$, that is, $\forall \pi_{*} \in \Pi_{*} \, \exists \pi \in \Pi_{\varepsilon} \, \mathbf{A}(\pi) \preceq \mathbf{c}(\pi_{*})$.

## BB-MO-CBS-pex

In this section, we first describe BB-MO-CBS-pex, a variant of BB-MO-CBS that computes an $\varepsilon$-approximate frontier for a given MO-MAPF problem instance and a user-provided $\varepsilon$-value. BB-MO-CBS-pex builds upon BB-MO-CBS-$\varepsilon$ with the two major improvements:

1. BB-MO-CBS-pex leverages A*pex to speed up the low-level search.
2. BB-MO-CBS-pex generalizes the merging idea of A*pex to reduce the sizes of joint paths for CT nodes (and hence speed up the high-level search).

In BB-MO-CBS-pex, each joint path $P$ represents a set of joint paths that are $\varepsilon$-dominated by $P$. BB-MO-CBS-pex maintains an *apex* for each joint path $P$ to keep track of the set of joint paths that were discarded due to being $\varepsilon$-dominated by $P$. Similar to an apex in A*pex, $\mathbf{A}(P)$ is the vector minimum of the costs of these joint paths. We say that $P$ is $\varepsilon$-*bounded* if $\mathbf{c}(P) \preceq_{\varepsilon} \mathbf{A}(P)$. When *merging* two joint paths $P$ and $P'$, the resulting joint path is either $P$ or $P'$, and the new apex is the vector minimum of the $\mathbf{A}(P)$ and $\mathbf{A}(P')$. BB-MO-CBS-pex merges two joint paths only when the resulting joint path is $\varepsilon$-bounded. Similar to BB-MO-CBS, a CT node in BB-MO-CBS-pex is a tuple $n = \langle \Omega, \{\Pi^i | i \in I\}, \mathbb{P} \rangle$, with two differences: (1) $\Pi^i$ for each agent $a^i$ is an $\varepsilon$-approximate frontier of paths that satisfy constraints in $\Omega$, and (2) $\mathbb{P}$ is a set of joint paths computed by merging joint paths in $PF(\Pi^1 \times \Pi^2 \times \cdots \times \Pi^{|I|})$. We use $\mathbb{P}.lexFirst$ to denote the joint path with the lexicographically smallest apex in $\mathbb{P}$ and call it the *current joint path* of node $n$. The $\mathbf{g}$-value of node $n$ is defined as $\mathbf{A}(\mathbb{P}.lexFirst)$.

Algorithm 2 shows the pseudocode for BB-MO-CBS-pex. We highlight the changes of BB-MO-CBS-pex over BB-MO-CBS using the blue text color. These changes are:

1. **Lines 3 and 24.** When initializing the root CT node $n_0$ and replanning paths for agents, BB-MO-CBS-pex uses A*pex to compute an $\varepsilon$-approximate frontier, instead of a Pareto frontier, of paths for each agent in each CT node.

2. **Lines 4 and 25.** When computing the set of joint paths for a CT node, BB-MO-CBS-pex calls $MergeJointPath$, which we will explain later, to compute an $\varepsilon$-approximate frontier of joint paths. This reduces the search effort of BB-MO-CBS-pex because fewer joint paths are considered for each CT node.

3. **Line 8.** When computing $\mathbb{P}'$ for an extracted CT node, BB-MO-CBS-pex calls $PruneApproxDom$ to remove a joint path $P$ if $\mathbf{A}(P)$ is $\varepsilon$-dominated by the cost of some solution $P_{sol}$ in $\mathcal{S}$. Additionally, BB-MO-CBS-pex updates $\mathbf{A}(P_{sol})$ to the vector minimum of $\mathbf{A}(P)$ and $\mathbf{A}(P_{sol})$ (Line 32). This update guarantees that, if BB-MO-CBS-pex merges $P_{sol}$ with other solutions later on Line 15, the cost of the resulting solution still $\varepsilon$-dominates $\mathbf{A}(P)$.

4. **Line 15.** When adding a solution $\mathbb{P}$ to $\mathcal{S}$, BB-MO-CBS-pex attempts to merge $\mathbb{P}$ with another solution in $\mathcal{S}$ on condition that the result solution is still $\varepsilon$-bounded. We show the merge function on Lines 40-43.

Algorithm 3 shows the pseudocode for function $MergeJointPath$. In $MergeJointPath$, BB-MO-CBS-pex iteratively computes $\mathbb{P}_i$, $i = 1, 2, \ldots, |I|$, where $\mathbb{P}_i$ is a set of joint paths for agent indices $a^1, a^2, \ldots, a^i$ and $\mathbb{P}_1$ is initialized with $\Pi^1$ (Line 1). To compute $\mathbb{P}_i, i = 2, 3, \ldots, |I|$, BB-MO-CBS-pex iterates over all combinations in $\mathbb{P}_{i-1} \times \Pi_i$, where each combination corresponds to a joint path $P$ for agents $\{a^1, a^2, \ldots, a^i\}$. BB-MO-CBS-pex first checks if $\mathbb{P}_i$ contains a joint path $P'$ that satisfies $Merge(P, P')$ is $\varepsilon$-bounded and, if so, replaces $P'$ (in $\mathbb{P}_i$) with $Merge(P, P')$(Line 8). Otherwise, $P$ is added to $\mathbb{P}_i$ (Line 9). Eventually, function $MergeJointPath$ returns $\mathbb{P}_{|I|}$.

We propose two additional improvement techniques:

**Choosing representative paths or joint paths based on conflicts:** While Zhang et al. (2022) proposed to choose representative paths based on the costs of paths for A*pex, we propose to chose representative paths (in the low-level search) or joint paths (in $MergeJointPath$) based on conflicts. In the low-level search, we use *Conflict Avoidance Tables* (CATs) (Sharon et al. 2015) to store the number of other agents passing via a given vertex or a given edge at a given timestep in the current joint path ($\mathbb{P}.lexFirst$). Therefore, for each path, we can compute its number of conflicts with other paths using the CAT. When merging two paths, the low-level search chooses the less conflicting path as the representative path on condition that the resulting node is $\varepsilon$-bounded and otherwise chooses the other path. In $MergeJointPath$, we also compute the number of conflicts for each joint path and prefer the less-conflicting joint path as the representative path when merging.

**Algorithm 2** BB-MO-CBS-pex

1: $\mathcal{S} \leftarrow \emptyset$; $Open \leftarrow \emptyset$
2: **for all** $i \in I$ **do**
3: $\quad \Pi_o^i \leftarrow ApproxLowLevelSearch(i, \emptyset, \boldsymbol{\varepsilon})$
4: $\mathbb{P}_o \leftarrow MergeJointPaths(\{\Pi_o^i | i \in I\}, \boldsymbol{\varepsilon})$
5: add $n_o = \langle \mathbb{P}_o, \emptyset, \{\Pi_o^i | i \in I\} \rangle$ to $Open$
6: **while** $Open \neq \emptyset$ **do**
7: $\quad n = \langle \mathbb{P}, \Omega, \{\Pi^i | i \in I\} \rangle \leftarrow Open.extract\_min()$
8: $\quad \mathbb{P}' \leftarrow PruneApproxDom(\mathbb{P})$
9: $\quad$ **if** $\mathbb{P}' = \emptyset$ **then continue**
10: $\quad$ **if** $\mathbb{P}'.lexFirst \neq \mathbb{P}.lexFirst$ **then**
11: $\quad\quad$ add $n = \langle \mathbb{P}', \Omega, \{\Pi^i | i \in I\} \rangle$ to $Open$
12: $\quad\quad$ **continue**
13: $\quad cft \leftarrow DetectConflict(\mathbb{P}'.lexFirst)$
14: $\quad$ **if** $cft = \emptyset$ **then**
15: $\quad\quad AddSolution(\mathbb{P}'.lexFirst)$
16: $\quad\quad$ remove $\mathbb{P}'.lexFirst$ from $\mathbb{P}'$
17: $\quad\quad$ **if** $\Pi \neq \emptyset$ **then**
18: $\quad\quad\quad$ add $\langle \mathbb{P}', \Omega, \{\Pi^i | i \in I\} \rangle$ to $Open$
19: $\quad\quad$ **continue**
20: $\quad \{\omega^i, \omega^j\} \leftarrow GenerateConstraints(cft)$
21: $\quad$ **for all** $i' \in \{i, j\}$ **do**
22: $\quad\quad \{\Pi_{\text{new}}^i | i \in I\} \leftarrow \{\Pi^i | i \in I\}$
23: $\quad\quad \Omega_{\text{new}} \leftarrow \Omega \cup \{\omega^{i'}\}$
24: $\quad\quad \Pi_{\text{new}}^{i'} \leftarrow ApproxLowLevelSearch(i', \Omega_{new}, \boldsymbol{\varepsilon})$
25: $\quad\quad \mathbb{P}_{\text{new}} \leftarrow MergeJointPaths(\{\Pi_{\text{new}}^i | i \in I\}, \boldsymbol{\varepsilon})$
26: $\quad\quad$ add $\langle \mathbb{P}_{\text{new}}, \Omega_{\text{new}}, \{\Pi_{\text{new}}^i | i \in I\} \rangle$ to $Open$
27: **return** $\mathcal{S}$
28: **procedure** $PruneApproxDom(\mathbb{P})$
29: $\quad \mathbb{P}' \leftarrow$ a copy of $\mathbb{P}$
30: $\quad$ **for all** $P \in \mathbb{P}'$ **do**
31: $\quad\quad$ **if** $\exists P_{sol} \in \mathcal{S}$ $\mathbf{c}(P_{sol}) \preceq_{\boldsymbol{\varepsilon}} \mathbf{A}(P)$ **then**
32: $\quad\quad\quad \mathbf{A}(P_{sol}) \leftarrow vector\_min(\mathbf{A}(P), \mathbf{A}(P_{sol}))$
33: $\quad\quad\quad$ remove $P$ from $\mathbb{P}'$
34: $\quad$ **return** $\mathbb{P}'$
35: **procedure** $AddSolution(P)$
36: $\quad$ **if** $\exists P_{\text{sol}} \in \mathcal{S}$ $Merge(P, P_{\text{sol}})$ is $\varepsilon$-bounded **then**
37: $\quad\quad$ replace $P_{\text{sol}}$ in $\mathcal{S}$ with $Merge(P, P_{\text{sol}})$
38: $\quad$ **else**
39: $\quad\quad$ add $P$ to $\mathcal{S}$
40: **procedure** $Merge(P, P')$
41: $\quad P_{\text{new}} \leftarrow$ choose from $P$ and $P'$
42: $\quad \mathbf{A}(P_{\text{new}}) \leftarrow vector\_min(\mathbf{A}(P), \mathbf{A}(P'))$
43: $\quad$ **return** $P_{\text{new}}$

**Algorithm 3** $MergeJointPaths$

**Input:** $\{\Pi^i \mid \forall i \in I\}, \boldsymbol{\varepsilon}$
1: $\mathbb{P}_1 \leftarrow \Pi^1$
2: **for all** $i = 2, 3, \ldots, |I|$ **do**
3: $\quad \mathbb{P}_i \leftarrow \emptyset$
4: $\quad$ **for all** $\langle P_{i-1} = \{\pi^1, \pi^2, \ldots, \pi^{i-1}\}, \pi^i \rangle \in \mathbb{P}_{i-1} \times \Pi^i$ **do**
5: $\quad\quad P \leftarrow \{\pi^1, \pi^2, \ldots, \pi^i\}$
6: $\quad\quad \mathbf{A}(P) \leftarrow \mathbf{A}(P_{i-1}) + \mathbf{A}(\pi^i)$
7: $\quad\quad$ **if** $\exists P' \in \mathbb{P}_i$ $Merge(P, P')$ is $\varepsilon$-bounded **then**
8: $\quad\quad\quad$ replace $P'$ with $Merge(P, P')$ in $\mathbb{P}_i$
9: $\quad\quad$ **else** add $P$ to $\mathbb{P}_i$
10: **return** $\mathbb{P}_{|I|}$

**Algorithm 4** $MergeJointPaths$, $AddSolutions$ and $MergeUntil$ for BB-MO-CBS-k

1: **procedure** $MergeJointPaths(\{\Pi^i : \forall i \in I\}, \boldsymbol{\varepsilon})$
2: $\quad \mathbb{P}_1 \leftarrow \{\{\pi^1\} \mid \pi^1 \in \Pi^1\}$
3: $\quad MergeUntil(\mathbb{P}_1, k)$
4: $\quad$ **for all** $i = 2, 3, \ldots, |I|$ **do**
5: $\quad\quad \mathbb{P}_i \leftarrow \emptyset$
6: $\quad\quad$ **for all** $\langle P_{i-1} = \{\pi^1, \pi^2, \ldots, \pi^{i-1}\}, \pi^i \rangle \in \mathbb{P}_{i-1} \times \Pi^i$ **do**
7: $\quad\quad\quad P \leftarrow \{\pi^1, \pi^2, \ldots, \pi^i\}$
8: $\quad\quad\quad \mathbf{A}(P) \leftarrow \mathbf{A}(P_{i-1}) + \mathbf{A}(\pi^i)$
9: $\quad\quad\quad$ add $P$ to $\mathbb{P}_i$
10: $\quad\quad MergeUntil(\mathbb{P}_i, k)$
11: $\quad$ **return** $\mathbb{P}_{|I|}$
12: **procedure** $AddSolution(P)$
13: $\quad$ add $P$ to $\mathcal{S}$
14: $\quad MergeUntil(\mathcal{S}, k)$
15: $\quad \varepsilon \leftarrow \max\{BF(P) \mid P \in \mathcal{S}\}$
16: **procedure** $MergeUntil(\mathbb{P}, k)$
17: $\quad$ **while** $|\mathbb{P}| > k$ **do**
18: $\quad\quad$ choose two joint paths $P$ and $P'$ from $\mathbb{P}$ such that $BF(Merge(P, P'))$ is minimized
19: $\quad\quad$ remove $P$ and $P'$ from $\mathbb{P}$
20: $\quad\quad$ add $Merge(P, P')$ to $\mathbb{P}$

**Eager solution update:** BB-MO-CBS and BB-MO-CBS-pex can be considered as updating solutions "lazily", that is, they try to update $\mathcal{S}$ only when extracting a node $n$ from $Open$ and the current joint path of $n$ is conflict-free.

We propose a eager solution-update scheme, which can be applied to both BB-MO-CBS($-\varepsilon$) and BB-MO-CBS-pex: In BB-MO-CBS-$\varepsilon$ with eager solution update, after Line 8 of Algorithm 1, we remove all conflict-free joint paths from $\mathbb{P}_{\text{new}}$, add these joint paths to $\mathcal{S}$, and remove dominated solution from $\mathcal{S}$. In BB-MO-CBS-pex with eager solution update, after Line 8 of Algorithm 2, we remove all conflict-free joint paths from $\mathbb{P}_{\text{new}}$ and call $AddSolution$ to add these joint paths to $\mathcal{S}$.

## BB-MO-CBS-k

In practice, it is unclear how to choose an appropriate $\boldsymbol{\varepsilon}$-value for a given MO-MAPF problem instance. If $\boldsymbol{\varepsilon}$ is set too large, BB-MO-CBS-$\varepsilon$ or BB-MO-CBS-pex might return only one solution, which provides no trade-off to users. If $\boldsymbol{\varepsilon}$ is set too small, BB-MO-CBS-$\varepsilon$ or BB-MO-CBS-pex might not benefit from approximation at all. Instead of specifying an approximation factor, one might prefer to specify a desirable number of solutions $k$. Therefore, we propose BB-MO-CBS-k, a variant of BB-MO-CBS-pex that computes a set of up to $k$ solutions for any user-specified $k$-value. In BB-MO-CBS-k, all components of $\boldsymbol{\varepsilon}$ are equal, i.e., $\boldsymbol{\varepsilon} = [\varepsilon, \varepsilon, \ldots, \varepsilon]$, and we will denote $\boldsymbol{\varepsilon}$ simply as $\varepsilon$ in the rest of this section.

BB-MO-CBS-k builds upon BB-MO-CBS-pex with the following changes:

1. The approximation factor $\varepsilon$ is initialized to zero and dynamically updated during the search.

2. Every time after low-level search for an agent $a^i$, BB-MO-CBS-k calls $MergeUtil$ to merge the set of paths

$\Pi^i$ until the size of $\Pi^i$ is no larger than $k$. $MergeUntil$ is explained later.

3. BB-MO-CBS-k uses a modified $MergeJointPaths$ function, which always outputs a set of at most $k$ joint paths, and a modified $AddSolutiuon$ function, which always keeps the size of $\mathcal{S}$ no larger than $k$. The modified $MergeJointPaths$ and $AddSolution$ are also explained later.

For a joint path $P$, we define its *boundedness factor* as

$$BF(P) := \max \left( 0, \max_{i=1,2,...,M} \frac{c_i(P)}{A_i(P)} - 1 \right).$$

It is easy to verify that $BF(P)$ is the smallest $\varepsilon$-value that satisfies joint path $P$ is $\varepsilon$-bounded (i.e., $c_i(P) \leq (1 + \varepsilon)A_i(P)$).

Algorithm 4 shows the pseudocode for $MergeUntil$ function, and modified $MergeJointPaths$ and $AddSolution$ functions. For a given set of joint paths (or a set of paths) $\mathbb{P}$, the $MergeUntil$ function iteratively chooses two joint paths $P$ and $P'$ from the input joint path set $\mathbb{P}$ such that $BF(Merge(P, P'))$ is minimized and replaces $P$ and $P'$ with $Merge(P, P')$ in $\mathbb{P}$ until the size of $\mathbb{P}$ is no larger than $k$. The modified $MergeJointPaths$ function for BB-MO-CBS-k calls $MergeUntil$ to keep the sizes of $\mathbb{P}_i$, $i = 1, 2, \ldots, |I|$, no larger than $k$ (Lines 3 and 10). The $AddSolution$ function for BB-MO-CBS-k also uses $MergeUntil$ to keep the size of $\mathcal{S}$ no larger than $k$. Additionally, it updates the approximation factor $\varepsilon$ to the largest bounded factor of $\mathcal{S}$. When BB-MO-CBS-k terminates, it returns $\mathcal{S}$, which contains no more than $k$ solutions. Additionally, $\mathcal{S}$ is guaranteed to be an $\varepsilon$-approximate frontier for the eventual value of $\varepsilon$.

## Theoretical Results

Due to the space limit, we put some of our proof in the appendix. Upon acceptance, we will purchase extra pages to include the complete proof or make our appendix available online.

**Definition (CVN set)** *Given a set of joint paths $\mathbb{P}$ and a node $n$ with constraints $\Omega$, let $CVN(n, \mathbb{P})$ be the set of all joint paths that (i) satisfy all constraints in $\Omega$, (ii) are conflict-free, and (iii) whose costs are not weakly dominated by the apex of any joint path in $\mathbb{P}$.*

We say a node $n$ *permits* a joint path $P$ with respect to $\mathbb{P}$ if $P \in CVN(n, \mathbb{P})$.

**Lemma 1.** *For agent $a^i$ and constraints $\Omega$, let $\Pi^i := ApproxLowLevelSearch(i, \Omega, \varepsilon)$. We have (1) for each path $\pi'$ of agent $a^i$ that satisfies $\Omega$, there exists a path $\pi \in \Pi^i$ with $\mathbf{A}(\pi) \preceq \mathbf{c}(\pi')$, and (2) all paths in $\Pi^i$ are $\varepsilon$-bounded.*

*Proof.* The lemma is shown by Theorem 1 in the paper of A*pex (Zhang et al. 2022). $\square$

**Lemma 2.** *Let $n_{\text{new}}$ denote node $\langle \mathbb{P}_{\text{new}}, \Omega_{\text{new}}, \{\Pi_{\text{new}}^i | i \in I\} \rangle$ that Algorithm 2 inserts to $Open$ on Line 26. We have (1) for any solution $P'$ that satisfies $\Omega_{\text{new}}$, there exists a joint path $P \in \mathbb{P}_{\text{new}}$ with $\mathbf{A}(P) \preceq \mathbf{c}(P')$ and (2) all joint paths in $\mathbb{P}_{\text{new}}$ are $\varepsilon$-bounded.*

BB-MO-CBS-pex uses $MergeJointPaths$ to compute $\mathbb{P}_{\text{new}}$. Therefore, to prove Lemma 2, we inductively show that, in Algorithm 3, for $i = 1, 2, \ldots, |I|$, $\mathbb{P}_i$ satisfy that for any conflict-free joint path $P'$ for agents $a^1, a^2, \ldots, a^i$ that satisfies $\Omega_{\text{new}}$, there exists a joint path $P \in \mathbb{P}_i$ with $\mathbf{A}(P) \preceq \mathbf{c}(P')$ and, all joint paths in $\mathbb{P}_i$ are $\varepsilon$-bounded. The complete proof is in the appendix.

**Lemma 3.** *When Algorithm 2 reaches Line 13, for any joint path $P \in CVN(n, \mathcal{S})$, there exists a joint path $P' \in \mathbb{P}'$ with $\mathbf{A}(P') \preceq \mathbf{c}(P)$.*

*Proof.* Before Algorithm 2 reaches Line 13, $n$ might have been previously extracted from and reinserted to $Open$ with different sets of joint paths. Let $\mathbb{P}_{\text{gen}}$ denote the set of joint paths computed by $MergeJointPaths$ when node $n$ was generated on Line 26. From Lemma 2, for any solution $P$ that satisfies $\Omega$, there exists a joint path $P' \in \mathbb{P}_{\text{gen}}$ with $\mathbf{A}(P') \preceq \mathbf{c}(P)$. Assume that $P'$ is in $\mathbb{P}_{\text{gen}}$ but not in $\mathbb{P}'$, which happens only if $P'$ has been removed on Lines 16 or 33. If $P'$ was removed on Lines 16, Algorithm 2 then added it to $\mathcal{S}$ on Line 15. If $P'$ was removed on Lines 33, the apex of some solution was updated to weakly dominate $\mathbf{A}(P')$ (Line 32). In both cases, there existed a solution in $\mathcal{S}$ whose apex weakly dominates $\mathbf{A}(P')$. Algorithm 2 might later merge this solution several (more) times with other solutions on Line 37 or update its apex on Line 32, but the apex of this solution will still weakly dominate $\mathbf{A}(P')$. We hence find a contradiction because, by the definition of CVN sets, the cost of $P$ is not weakly dominated by the apex of any solution in $\mathcal{S}$. $\square$

**Lemma 4.** *At the beginning of each iteration of BB-MO-CBS-pex (i.e., before executing Line 7), for any solution $P$, if there does not exist a solution $P_{sol} \in \mathcal{S}$ with $\mathbf{A}(P_{sol}) \preceq \mathbf{c}(P)$, there exists a node $n \in Open$, which permits $P$ with respect to $\mathcal{S}$.*

**Theorem 1.** *Given an MO-MAPF instance that has at least one solution, when BB-MO-CBS-pex terminates, $\mathcal{S}$ is an $\varepsilon$-approximate frontier.*

**Theorem 2.** *Given an MO-MAPF instance that has at least one solution, BB-MO-CBS-pex terminates in finite time.*

## Experimental Results

In our experimental results, we evaluated (1) BB-MO-CBS, (2) BB-MO-CBS-$\varepsilon$, (3) BB-MO-CBS-pex, (4) BB-MO-CBS-pex-E (BB-MO-CBS-pex with eager-solution update), (5) BB-MO-CBS-pex-E-CB (BB-MO-CBS-pex-E with conflict-based merging), and (6) BB-MO-CBS-k. All algorithms are implemented in C++[1] and share a common base as much as possible. We conducted all experiments on a Ubuntu 20.04.5 laptop with an Intel Core i7-10510U 1.80GHz CPU and 16GB RAM.

The low level of BB-MO-CBS and BB-MO-CBS-$\varepsilon$ is implemented with BOA* for bi-objective domains and

---

[1]Upon acceptance, the code and the data will be made publicly available.

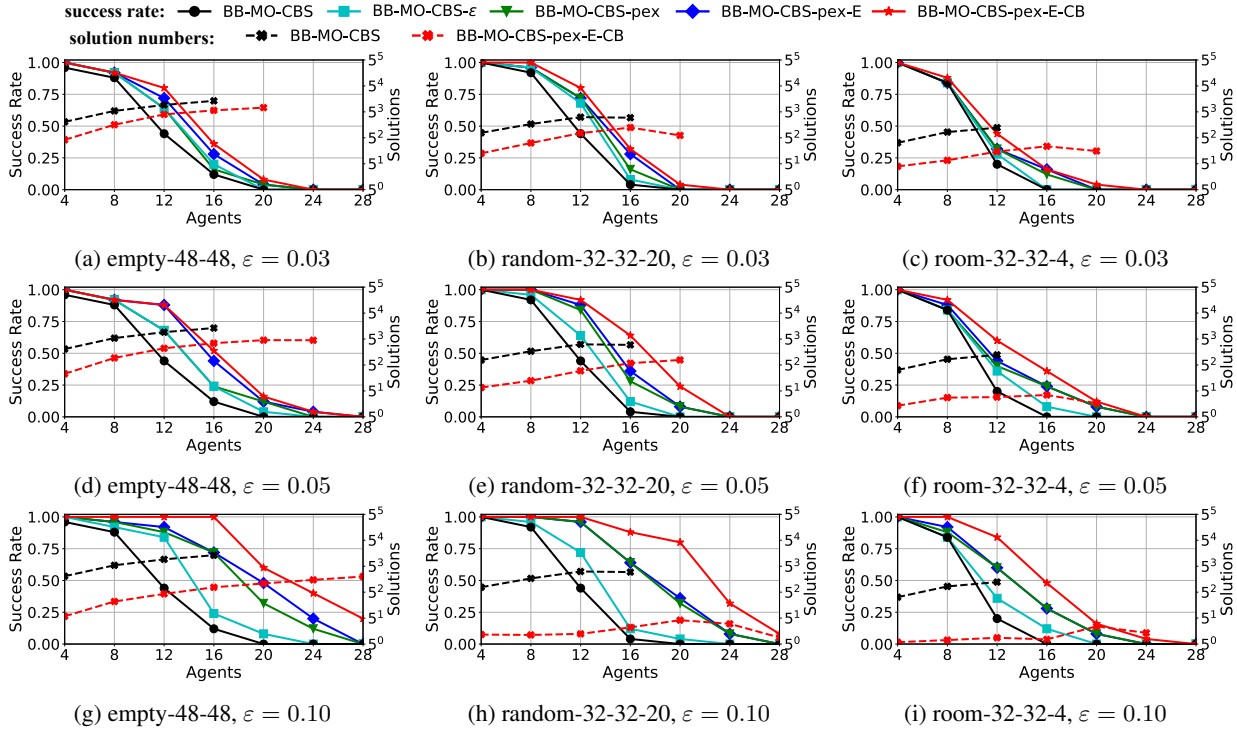

Figure 2: Experimental results for bi-objective instances.

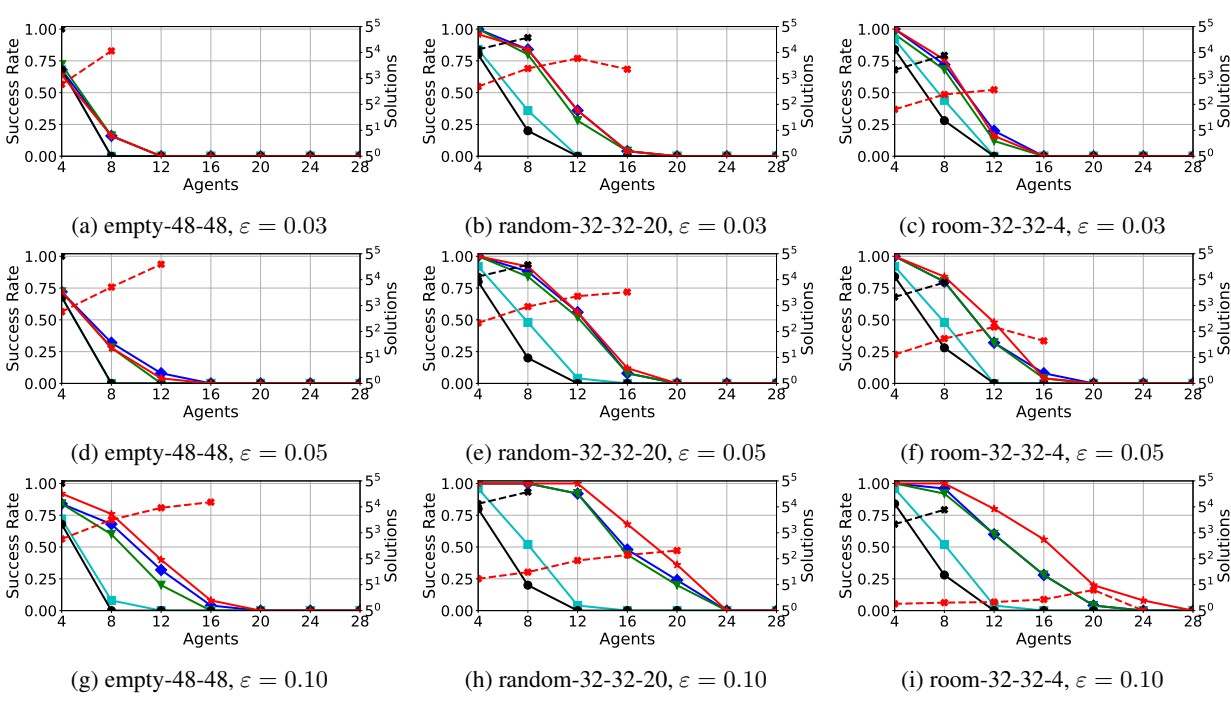

Figure 3: Experimental results for tri-objective instances.

NAMOA*dr for domains with more than two objectives. The BB-MO-CBS-pex variants without conflict-based merging use the "greedy" merging strategy, which is proposed by Zhang et al. (2022) and has the best overall performance among different merging strategies.

mance among different merging strategies.

We use three four-connected grids from the MAPF benchmark (Stern et al. 2019): empty-48-48, random-32-32-20, and room-32-32-4. We generate the cost for each edge by

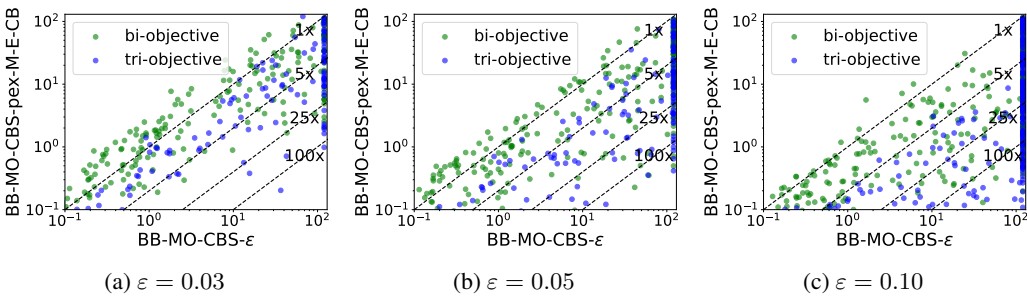

(a) $\varepsilon = 0.03$       (b) $\varepsilon = 0.05$       (c) $\varepsilon = 0.10$

Figure 4: Runtime results for BB-MO-CBS-$\varepsilon$ and BB-MO-CBS-E-CB

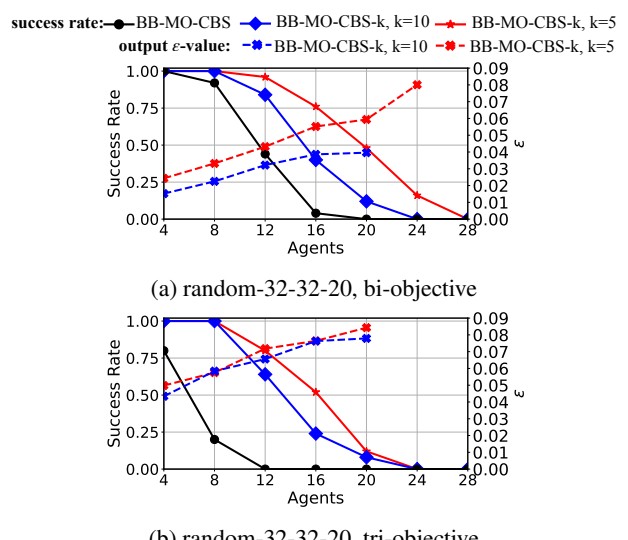

(a) random-32-32-20, bi-objective

(b) random-32-32-20, tri-objective

Figure 5: Experiment results for BB-MO-CBS and BB-MO-CBS-k

randomly sampling each cost component from 1 to 5. The MAPF benchmark contains 25 random scenarios for each map, and each scenario provides a list of start-goal pairs. For each scenario, we vary the number of agents $N$ from 4 to 28 and generate problem instances with the first $N$ start-goal pairs. We run experiments with two and three objectives and a runtime limit of 120 seconds for each problem instance.

### Different variants of BB-MO-CBS-pex

We compare BB-MO-CBS, BB-MO-CBS-$\varepsilon$, and different variants of BB-MO-CBS-pex on empty-48-48, random-32-32-20, and room-32-32-4 with approximation factors of 0.03, 0.05, and 0.1.

Figures 2 and 3 show the experimental results for different algorithms on instances with two and three objectives, respectively. The solid lines show the success rate (i.e., the percentage of instances solved by an algorithm within the runtime limit) for each algorithm. BB-MO-CBS-$\varepsilon$ has higher success rates than BB-MO-CBS. BB-MO-CBS-E has higher success rates than BB-MO-CBS-$\varepsilon$, while, in turn, BB-MO-CBS-E-CB has significantly higher success rates than BB-MO-CBS-pex-E, which shows the usefulness of the eager-

solution update and conflict-based merging techniques. The improvements in success rates of these techniques are more significant for $\varepsilon$-values of 0.05 and 0.10. For example, in random-32-32-30 with two objectives and 20 agents, the addition of conflict-based merging doubles the success rate. The dashed lines in Figures 2 and 3 show the average numbers of solutions of BB-MO-CBS and BB-MO-CBS-pex-M-CB. We can see that introducing approximation to the MO-MAPF problem reduces the sizes of solution sets by up to two orders of magnitude.

Figure 4 shows the individual runtime of BB-MO-CBS-$\varepsilon$ and BB-MO-CBS-E-CB for each problem instance. BB-MO-CBS-pex-M-CB takes significantly less time than BB-MO-CBS-$\varepsilon$ in general. When $\varepsilon$ becomes larger, and on instances with three objectives, BB-MO-CBS-pex-M-CB becomes significantly more efficient than BB-MO-CBS-$\varepsilon$.

### BB-MO-CBS-k

We compare BB-MO-CBS and BB-MO-CBS-k on random-32-32-20 with two and three objectives. For BB-MO-CBS-k, we use two $k$-values, 5 and 10. The experimental results are shown in Figure 5. The solid lines show the success rate for each algorithm, and we can see that BB-MO-CBS-k has significantly higher success rates than BB-MO-CBS. The dashed lines show the average approximation factor output by BB-MO-CBS-k, and we can see that, with $k = 5$ and $k = 10$, BB-MO-CBS-k still computes solution sets with approximation factors smaller than 0.1.

## Conclusions

In this paper, we proposed BB-MO-CBS-pex, which leverages A*pex to compute approximate frontiers for MO-MAPF problems for user-specified approximation factors. Based on BB-MO-CBS-pex, we proposed BB-MO-CBS-k, which computes up to $k$ solutions for a user-provided $k$-value. Our experimental results show that both BB-MO-CBS-pex and BB-MO-CBS-k solved significantly more instances than BB-MO-CBS for different approximation factors and $k$-values, respectively. We also show that BB-MO-CBS-pex achieved speed-ups up to two orders of magnitude compared to BB-MO-CBS-$\varepsilon$, our baseline approximation variant of BB-MO-CBS.

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
