# OpenReview forum: "Efficient Approximate Search for Multi-Objective Multi-Agent Path Finding"
_icaps-conference.org/ICAPS/2024/Conference — ICAPS 2024_

### Official Review · Reviewer_6goQ · 2023-12-25

**Significance And Importance:** 2
**Soundness:** 3
**Novelty:** 2
**Clarity:** 3
**Confidence:** 5

**Weaknesses:**

1: Minor weaknesses that are easily fixable.

**Contributions Of The Paper:**

This paper delves into two extensively researched problems: the Multi-Agent Path Finding (MAPF) problem, which has received significant attention in the last decade, and the multi-objective shortest path (MOSP) problem, which has been under study for over two decades and has recently regained interest.

**Ethical Considerations:**

(1) Not Applicable: The paper does not have any ethical considerations to address

**Nomination For Best Paper:**

No

**Overall Evaluation:**

-1: (weak reject)

**Questions For Authors:**

What can you say about howwell can the k-solution approximatethe PF? Can this be appliedto the single-agent setting?

**Reproducibility:**

4: Authors promise to release code and domains (whichever apply).

**Strengths Of The Paper:**

The study introduces an approach aimed at approximating the Pareto frontier (PF) by extending A*pex, a recent method focusing on approximating the PF for the single-agent MOSP. While the paper's overall structure seems robust and devoid of major flaws, the technical sections might pose readability challenges. The empirical findings, impressive in terms of speed enhancements, were somewhat expected given the relatively basic nature of the baseline methods. This limitation arises due to the absence of more sophisticated benchmarks specifically designed to compute the approximate PF for MOS-MAPF problems.

**Weaknesses Of The Paper:**

The paper amalgamates two concepts or algorithms: A*pex and BB-MO-CBS, the latter being an adaptation of CBS (a MAPF solver) to the MOSP setting. While the combination does not pose significant challenges, two ideas stand out beyond mere amalgamation:
(*) I really liked the idea of choosing representative paths or joint paths based on conflicts since this is intimately related to the MAPF part of the problem.
(*) The concept of k-solutions, although seemingly detached from the MAPF aspect, holds independent interest within the (single-agent) MOSP community. However, it remains unclear whether these k solutions offer any approximation of the PF. Additionally, the evaluation benchmarks used for these solutions were notably simplistic. A more fitting baseline approach could involve running BB-MO-CBS-pex with decreasing epsilon values until reaching k.

The authors motivate their work by stating that "Many real-world applications of MAPF can be viewed as multi-objective problems. For example, in multi-robot systems, some interesting cost metrics are travel distance, energy consumption, and risk." The same sentence could have been written for the single-agent setting. In the MAPF literature, researchers typically consider two cost functions - the Sum of Costs (SOC) and the Makespan which are related but may differ. There is no clear case when each should be used and so combining the two in the context of MOS is the most natural step if MOS is considered. This is not trivial since Makespan is not additive over edges but I am sure that this is a technicality that can be addressed. It is unfortunate that these interesting and more relevant questions were overlooked.

Finally, small issues
(*) Bottom of page 5
	- "after low-level search" => "after \insert{the} low-level search"
	- MergeUtil => MergeU\insert{n}til
(*) References:
	- 	Ren, Z.; Rathinam, S.; and Choset, H. 2022.
		A Conflict-Based Search Framework for Multi-Objective Multi-Agent Path Finding.
		https://arxiv.org/abs/2108.00745.

		Replace with journal paper:

		Zhongqiang Ren, Sivakumar Rathinam, Howie Choset:
		A Conflict-Based Search Framework for Multiobjective Multiagent Path Finding.
		IEEE Trans Autom. Sci. Eng. 20(2): 1262-1274 (2023)

	- 	Ulloa, C. H.; Yeoh, W.; Baier, J. A.; Zhang, H.; Suazo, L.; and Koenig, S. 2020.
		A Simple and Fast Bi-Objective Search Algorithm.
		In Proceedings of the International Conference on Automated Planning and Scheduling, volume 30, 143–151.

		Replace with journal paper:

		Carlos Hernández, William Yeoh, Jorge A. Baier, Han Zhang, Luis Suazo, Sven Koenig, Oren Salzman:
		Simple and efficient bi-objective search algorithms via fast dominance checks.
		Artif. Intell. 314: 103807 (2023)

---

> ### Author Rebuttal · Authors · 2024-01-28
>
> Thank you for your feedback.
> 1. “it remains unclear whether these k solutions offer any approximation of the PF.” -Different from BB-MO-CBS-pex, where ε-value is specified as an input argument, BB-MO-CBS-k dynamically updates its ε-value during the search. When BB-MO-CBS-k terminates, the solution set is “guaranteed to be an ε-approximate frontier for the eventual value of ε.” However, BB-MO-CBS-k does not have any guarantee on what the eventual value of ε will be. Empirically, in the problem instances we used, the eventual ε-values are always within 0.1, as shown in Fig. 5. We will add more explanation in the final version to clarify this.
>
> 2. “A more fitting baseline approach could involve running BB-MO-CBS-pex with decreasing epsilon values until reaching k.” -Running BB-MO-CBS-pex with decreasing epsilon values could be an interesting direction for future work. We think the main question is how to specify the sequence of decreasing epsilon values.
>
> 3. “In the MAPF literature, researchers typically consider two cost functions - the Sum of Costs (SOC) and the Makespan which are related but may differ.” -It is an interesting cost metric. We think makespan is hard to optimize at the low level, however, the high-level merging strategy can be adjusted to optimize makespan. The details are worth exploring in future work.
>
> 5. “Can this be applied to the single-agent setting?” -The main changes on BB-MO-CBS-k over BB-MO-CBS-pex are on the high level, and hence might not be directly applicable to the single-agent setting. However, this can definitely be interesting future work.
>
> We will also address all the grammar issues that you pointed out.

---

### Official Review · Reviewer_4vpQ · 2024-01-10

**Significance And Importance:** 2
**Soundness:** 3
**Novelty:** 2
**Clarity:** 3
**Overall Evaluation:** 1
**Confidence:** 2

**Weaknesses:**

1: Minor weaknesses that are easily fixable.

**Contributions Of The Paper:**

This paper proposes to extend the BB-MO-CBS approach for MO-MAPF. Firstly, it leverages A*pex, a single agent MO path finding algorithm, to compute MO single agent paths that are included in the set of joint paths to then epsilon-approximate the solution set. The proposed BB-MO-CBS-pex algorithm iteratively solves conflicts using a conflict-based tree search - where each node of the tree is represented by a set of the constraints, the set of paths single agent paths and the set of joint paths – until a epsilon-approximated Pareto solution is found. Authors also propose two variants of BB-MO-CBS-pex algorithm : one that chooses representative paths based on conflicts and another one that updates the solution set in a greedy fashion. Additionally, the authors propose another variant for BB-MO-CBS-pex where the user is asked to define how many solutions he/she wants (e.g. k solutions representing the epsilon-aproximate Pareto frontier), called BB-MO-CBS-k. This variant computes dynamically the epsilon factor during iterations and gives k joint paths solutions. A theoretical analysis is presented and the experiments suggest the approach is sound and more efficient then the state-of-the-art.

**Ethical Considerations:**

(1) Not Applicable: The paper does not have any ethical considerations to address

**Nomination For Best Paper:**

No

**Questions For Authors:**

- Could authors comment more on the benefits on BB-MO-CBS-pex(-E)-CB in terms of number of conflicts. Are the paths proposed by this variant "less conflicting" when compared with the others ?
- If I well understood, BB-MO-CBS-k is based on BB-MO-CBS-pex and not on BB-MO-CBS-pex-E-CB (the more effective approach), why ?
- Why the authors did not compared  BB-MO-CBS-pex(-E-CB) with  BB-MO-CBS-k ?

**Reproducibility:**

4: Authors promise to release code and domains (whichever apply).

**Strengths Of The Paper:**

This paper contributes with some new ideas (k solutions for instance) and represents, in my opinion, several incremental advances with respect to the state-of-the-art approach it is based on.
The paper is clear and well-organized despite the complex ideas behind the contributions.
The theoretical analysis is rich, and authors promise to include the missing lemmas/theorems proofs on the final version (by purchasing additional pages (?)).
The results on three benchmarks of the literature suggest the approach is effective.

**Weaknesses Of The Paper:**

I did not identified any major weakness in the paper. However some minor details could be improved.

Eager solution update : in Line 8 of Alg.1 (resp Alg. 2) refers to P’ not P_new.

I suggest to the authors to add some information on the experimental section. For instance, which are the cost components used in experiments.

Additionally, y-axis labels and legend of figures 4.a, 4.b and 4.c and the text in paragraph explaining Fig. 4 results should be also corrected (e.g. BB-MO-CBS-pex-M-E-CB →  BB-MO-CBS-pex-E-CB). And, in last lines in paragraph « Figures 2 a 3 [...] »,  BB-MO-CBS-E-CB should be corrected to BB-MO-CBS-pex-E-CB.

Significantly higher success rates : authors argue that the success rate obtained with  BB-MO-CBS-pex-E-CB is significantly higher then  BB-MO-CBS-pex-E, but these results are not supported by statistical analysis. So, the term « significantly » should not be used in my opinion.

---

> ### Author Rebuttal · Authors · 2024-01-28
>
> Thank you for your feedback.
>
> 1. "I suggest to the authors to add some information on the experimental section. For instance, which are the cost components used in experiments." -We provided the information on how the costs are generated in the paragraph below Figure 5: "We generate the cost for each edge by randomly sampling each cost component from 1 to 5." We will also release the data set and code we used.
>
> 2. "Significantly higher success rates: authors argue that the success rate obtained with BB-MO-CBS-pex-E-CB is significantly higher than BB-MO-CBS-pex-E, but these results are not supported by statistical analysis." -We will rephrase "BB-MO-CBS-E-CB has significantly higher success rates than BB-MO-CBS-pex-E" to "BB-MO-CBS-E-CB has higher success rates than BB-MO-CBS-pex-E in almost all cases."
>
> 3. "Could authors comment more on the benefits of BB-MO-CBS-pex(-E)-CB in terms of the number of conflicts?" -The solutions computed by BB-MO-CBS-pex (with or without CB) are always conflict-free. Yet, during the search, CB tends to propose “less conflicting” paths. Specifically, with CB, the low-level planner chooses representative paths based on the number of conflicts with the paths of other agents in the current joint path, and hence, the replanned paths are more likely to have fewer conflicts with other agents than those computed without CB. With CB, the high level (in MergeJointPath), merges two joint paths based on the number of conflicts in each joint path. Thus, the resulting joint path is more likely to be less conflicting.
>
> 5. "BB-MO-CBS-k is based on BB-MO-CBS-pex and not on BB-MO-CBS-pex-E-CB (the more effective approach), why?" -Our implementation of BB-MO-CBS-k also uses the (-E) and (-CB) techniques. We will make this clear in the final version.
>
> 6. "Why the authors did not compare BB-MO-CBS-pex(-E-CB) with BB-MO-CBS-k?" -BB-MO-CBS-pex and BB-MO-CBS-k are based on two different paradigms. BB-MO-CBS-pex computes an $\varepsilon$-approximate PF for a user-provided $\varepsilon$-value, while BB-MO-CBS-k computes $k$-solutions for a user-provided $k$-value. They are useful for different cases. We couldn’t determine these two hyperparameters to make a meaningful comparison.
>
> We will also address all the grammar issues that you pointed out.

---

### Official Review · Reviewer_5u6A · 2024-01-21

**Significance And Importance:** 2
**Soundness:** 3
**Novelty:** 3
**Clarity:** 3
**Overall Evaluation:** 1
**Confidence:** 3

**Weaknesses:**

0: Minor weaknesses requiring some work to be addressed for the paper to be accepted.

**Contributions Of The Paper:**

The paper applies A*pex, which is a multi-objective single-agent search algorithm that computes ε-approximate frontier as a set of solutions, to BB-MO-CBS, which is a multi-objective multi-agent search derived from CBS that does not use approximation, resulting in BB-MO-CBS-pex, a multi-objective multi-agent search algorithm that computes an ε-bounded Pareto frontier of the solution where the value of ε is user-specified.
Becauase it may be inconvenient to have the user decides on the value of ε, the paper also proposes BB-MO-CBS-k, a variant of BB-MO-CBS-pex that computes a set of up to k solutions for any user-specified k-value.
The paper also proposes two techniques to speed up BB-MO-CBS-pex and BB-MO-CBS-k: (1) conflict-based merging (i.e., choosing the less conflicting path as the representative path) and (2) eager solution update, with the aim to further reduce the search space.

**Ethical Considerations:**

(1) Not Applicable: The paper does not have any ethical considerations to address

**Nomination For Best Paper:**

No

**Questions For Authors:**

Page 5: "In BB-MO-CBS-ε with eager solution update, after Line 8 of Algorithm 1, we remove all conflict-free joint paths from P_{new}" -> Do you mean P'?
Alg 1 Line 7: "n = (...)" -> "n = <...>"?
Alg 1 Line 8: Remove the "∧" to keep the writting style consistent?
Alg 2, Lemma 2: Shouldn't the notion of a CT node be written as <Omega, {...}, P> as defined in the introduction of BB-MO-CBS-pex?
Page 5: "MergeUtil" -> "MergeUntil".
Figs 2, 3: "solution numbers" -> "number of solutions"?
Alg 4: "AddSolutions" -> "AddSolution".
Page 5: "propose a eager" -> "propose an eager".
Page 6: "AddSolutiuon" -> "AddSolution".
Fig 4: "BB-MO-CBS-pex-M-E-CB" -> "BB-MO-CBS-pex-E-CB"?
Page 8: Three instances of "MO-CBS-pex-M-CB" -> "MO-CBS-pex-E-CB"?

**Reproducibility:**

4: Authors promise to release code and domains (whichever apply).

**Strengths Of The Paper:**

The paper proposes an advancement over a state-of-the-art MO-MAPF algorithm, allowing it to solve more problems. The tradeoff is lower-quality solutions, but they are still bounded by ε which is user-specified.

**Weaknesses Of The Paper:**

I find the proof somewhat difficult to follow. This may be because I still need more time or a complete proof is not given due to space limitation. The authors promise to purchases extra pages to put the complete proof upon acceptance.

Post rebuttal: Thank you for providing the rebuttals. I did lower my score a bit due to the fact that it appears that a better experimental evaluation would be needed to highlight the salient features of the proposed algorithms.

---

> ### Author Rebuttal · Authors · 2024-01-28
>
> Thank you for your feedback.
> 1. “I find the proof somewhat difficult to follow. This may be because I still need more time or a complete proof is not given due to space limitation. The authors promise to purchases extra pages to put the complete proof upon acceptance.”
> Yes, we had to keep the proof succinct due to the space limit. In the final version, we will improve the clarity of the proof with more explanation and buy extra pages for the complete proof.
>
> We will also address all the grammar issues that you pointed out.

---

### Meta-Review · Area_Chair_DZn9 · 2024-02-05

**Recommendation:** Accept (Oral)
**Confidence:** 2

**Metareview:**

This paper lead to a lot of discussion. The scores were at first in general positive (+1/+1/+2) and the reviews deemed it an advancement of the state of the art. On the other hand, the rebuttal was not received well due to a lack of detail and the authors seemingly not intending to improve the paper based on the reviews. For example, two reviewers were dissatisfied with the lack of comparison between the $\epsilon$-approximate algorithm and the $k$-approximate algorithm, which the authors agreed would be interesting but deferred to future work. This resulted in two reviews lowering their score (2 -> 1 and 1 -> -1). Another concern was that a proof lacked important details, but this was resolved by the authors promising to move the more detailed proof from the supplementary material to the main paper by buying additional pages.

While the scores overall are positive, it is noteworthy that the only negative score comes from the reviewer with the highest confidence. Nevertheless I weakly recommend to accept the paper due to the following:
1) The reviews were at first positive.
2) The paper is both theoretically sound and performs well empirically.
3) If I were the author it would not have been clear to me from the initial reviews that the positive score is depending on revising and extending parts of the paper.

**Ethical Considerations:**

(1) Not Applicable: The paper does not have any ethical considerations to address